# A Very Large Spawning Aggregation of a Deep-Sea Eel: Magnitude and Status

Alan Williams [1,*], Deborah Osterhage [2], Franziska Althaus [1], Timothy Ryan [1], Mark Green [1] and John Pogonoski [1]

1 CSIRO Oceans and Atmosphere, Marine Laboratories, Castray Esplanade, Hobart, TAS 7001, Australia; franzis.althaus@csiro.au (F.A.); Tim.ryan@csiro.au (T.R.); mark.green@csiro.au (M.G.); john.pogonoski@csiro.au (J.P.)
2 Institute for Marine and Antarctic Studies, University of Tasmania, Hobart, TAS 7001, Australia; deboraho@utas.edu.au
* Correspondence: alan.williams@csiro.au

**Abstract:** Multiple lines of evidence substantiate the existence of a very large aggregation of the basketwork eel, *Diastobranchus capensis*, on the small (3 km$^2$) Patience Seamount off southeast Australia. The aggregation appears to be present year-round, but largest in the austral autumn when composed of spawning eels. Twenty eels caught in April 2015 (14 female, 6 male) were all in advanced stages of spawning condition. The eel's abundance in the aggregation was very high as measured at seamount, local and regional scales. Hydroacoustic measurement of the spawning aggregation's dimensions (~100 × 1000 m) and conservative counts of 100 s of eels along camera transects of ~1000–2000 m length indicate 10,000 s individual eels may have been present. The absence of other known spawning locations indicates the Patience Seamount is a regional-scale spatial anchor for spawning. The aggregation was protected in a marine park in 2007 following a decades-long impact from bottom trawling, indicating that the population can be expected to stabilise and recover. Monitoring the aggregation's status, and validating seasonal spawning, provide important opportunities to examine conservation-led recovery in the deep sea as part of Australia's new national strategy of Monitoring, Evaluation, Reporting and Improvement (MERI) for conservation values within marine parks.

**Keywords:** *Diastobranchus capensis*; seamount; trawling impact; marine park; monitoring

## 1. Introduction

Over 1000 species of true eels frequent world oceans [1] from coastal waters to hadal depths [2], but little is known of the spawning ecology of eels. The most studied are the catadromous anguillid eels, known to migrate and spawn in the oceans. Approximate spawning locations, based largely on leptocephalus catches, have been estimated for all temperate and most tropical anguillid species, with the first described over 100 years ago [3]. Intensive search efforts over the last two decades using increasingly sophisticated technology have resulted in better information for spawning locations, including capture of some mature adult eels, but have not revealed any direct observations of spawning aggregations [3–5]. Marine eels are believed to have a number of spawning strategies indicated by leptocephalus catch, namely long migrations to offshore spawning areas, short migrations to the shelf break or deep water near the continental shelf, or no migration [6]. There are few reported instances of marine eel spawning, and most are inshore including observations of spawning moray and garden eels [7–9], and a suggested spawning aggregation based on increased catch of mature conger eels within a harbour [10]. Aggregations of marine eels are also rarely reported: ophichthid eels have been seen aggregating offshore at the sea surface (n ≥ 62), but not spawning [11,12], and there have been two observations of synaphobranchid eel aggregations on seamounts, both associated with hydrothermal vents, namely *Dysommina rugosa* (unquantified) [13]; and *Maedia abyssalis* (n = 30–40) [14].

The latter was speculated to be a spawning aggregation because the eels' skin condition resembled that of *Anguilla* eels that had been experimentally induced to spawn. Eels are also known to be attracted in numbers to baited cameras, with the most significant being the synaphobranchis eel *Ilyophis arx* on an abyssal seamount (max N = 115) [15]. To date, however, there have been no observations of confirmed spawning aggregations.

In this article, we provide evidence for an oceanic spawning aggregation of the basketwork eel *Diastobranchus capensis* (Synaphobranchidae), a large-bodied, deep-sea eel found in mid-continental slope (~700 to ~2000 m depths). The species has a very broad distribution, bounded in the south by the Antarctic Convergence [16,17], and extending into the North West Atlantic [18], and Arctic [19]. However, its core distribution is Southern Hemisphere where it is near circum-austral, and most abundant off temperate Australia and around New Zealand, including on seamounts [20,21].

Little is known of the ecology and biology of *D. capensis* despite it being a dominant mid-slope species. The species occurs frequently at low densities in scientific and commercial bottom trawl catches [20,22,23] and lone individuals are often observed by towed cameras (authors' observations). Most ecological studies have focussed on its trophic status: the eel is predominantly a piscivore [24], although wider, predominantly mesopelagic, prey ranges are also reported [25,26], and a facultative scavenger [27]; it can also be attracted to baited videos where it has been seen in groups of up to 11 [28]. Specific knowledge of its reproductive ecology is virtually non-existent. Only one gravid specimen had been previously reported [29,30] but no details of the gonad and leptocephalus have been specifically described.

Using opportunistically collected samples from three surveys, our overall objective was to characterize a biological aggregation on Patience Seamount south of Tasmania, Australia which was anecdotally reported as being composed of the deep-sea eel, *Diastobranchus capensis*. Specifically, we aimed to: (1) map and characterize the aggregation and identify its composition using photographic imagery, net catches and hydro-acoustics; (2) if substantiated as an aggregation of *D. capensis*, establish its uniqueness by comparing the species' abundance on adjacent seamounts and with fishery catch data from its core range in Australia and New Zealand; and (3) investigate the aggregation as a spawning aggregation based on reproductive data of *D. capensis* collected within the aggregation, and infer its seasonality based on data taken at two times during the year (March/April and December). Finally, we discuss the aggregation in terms of its ecological value—from the standpoints of its ecological role, and its recovery from fishing impacts following protection within the 'Huon' Australian Marine Park [31]. We make recommendations for future monitoring of the aggregation in the context of management planning for the marine park.

## 2. Materials and Methods

### 2.1. Field Sampling

Samples and data from the eel aggregation were collected opportunistically during three surveys that aimed primarily to map the seabed and associated benthic biodiversity on the continental slope off southern Tasmania (Figure 1) in an area containing a cluster of small volcanic seamounts see [32]: survey #1 in March/April 2007 (RV *Southern Surveyor*, SS200702); #2 in April 2015 (RV Investigator IN2015_E02; and #3 in November/December 2018 (RV *Investigator* IN2018_V06). A deep-water towed camera platform [33] was used on all surveys to acquire imagery that permitted the relative abundance and density of *D. Capensis* and co-occurring fishes to be estimated on 10 other adjacent seamounts (Table 1). Real-time video feed through a fibre-optic cable enabled the camera to be maintained at a consistent height (2 to 4 m) off the seabed where it provided continuous stereo-video and regular (5 s interval) high-resolution still images from cameras with an oblique field of view (~5 m wide by 5.5 m depth of field). The camera's geo-location was estimated to an accuracy within 20 m by cross-referencing the GPS location of the ship with an ultra-short baseline (USBL) tracking beacon mounted on the camera platform; further details are given

in [32]. Specimens of *D. capensis* were captured by a small (4 m wide) beam trawl [34] during survey #2 and with a baited-hook and line on survey #3.

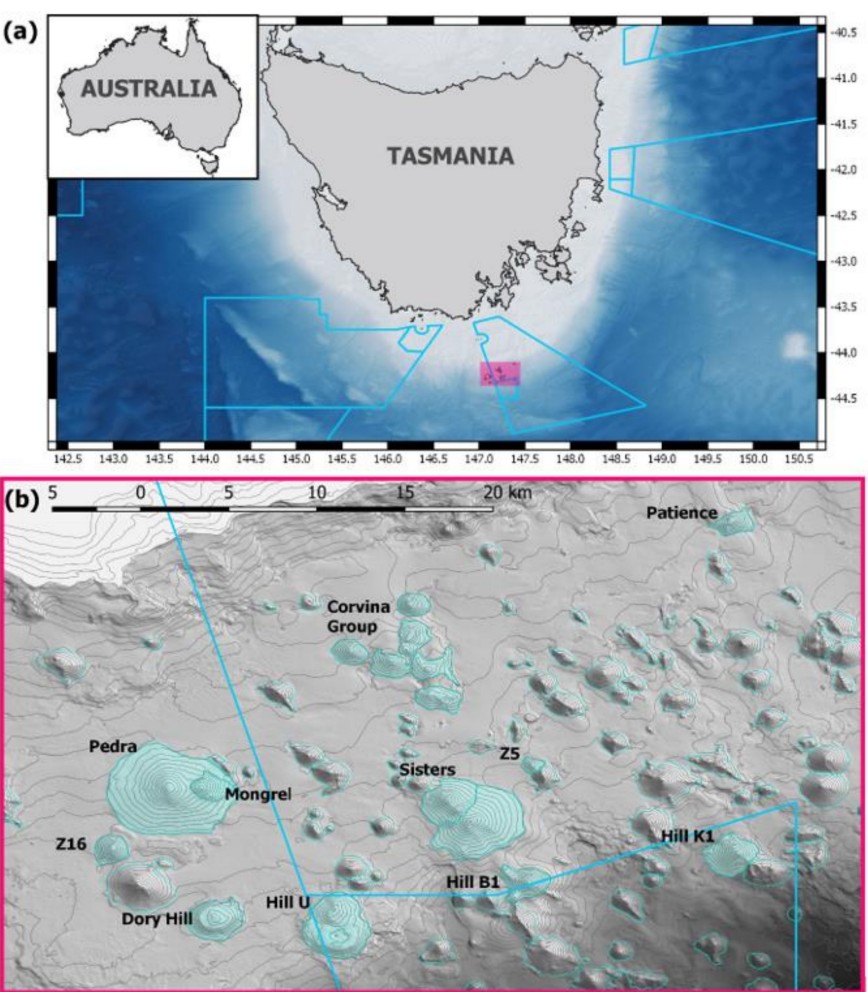

**Figure 1.** Study area: (**a**) Location of the study area (red shaded area) south of Tasmania (inset: Australia) with the boundaries of Australian Marine Parks (AMP) shown in cyan; (**b**) multibeam sonar bathymetry map of the seamount cluster in the study area with 50 m depth contours overlaid in ~200–2000 m depths; the sampled seamounts are highlighted and named; cyan lines are parts of the boundary of the 'Huon' AMP.

### 2.2. Acoustic Mapping of the Aggregation at Patience Seamount

The 'Patience Seamount' is a small (3 sq km base area) extinct volcano, which was mapped for the first time during survey #1 using a 30 KHz Simrad EM300 multibeam echosounder. This seamount lies at the shallow margin of a cluster of similarly conical-shaped and mostly small (1 to 10, max 25 sq km base area) seamounts, all of which intersect the depth range occupied by *D. capensis* (approximately 700–2000 m) (Figure 1b). During survey #2, two sets of 'star-pattern' transects were made with a calibrated EK60 single beam echosounder at 18 and 38 KHz frequencies during the daytime: set #1 on the 10 April 2015 at 21:53 UTC (07:53 local) and set #2 at 04:17 UTC (14:07 local) to identify the extent and dynamics of the eel aggregation. Additional echosoundings were collected on 12 April during photographic transects. Technical problems lead to the loss of raw acoustic data on 10 April, but images of the real-time display of the Simrad EK60 38 kHz echosounder taken from the Simrad ER60 acquisition software (version 2.4.3, accessed on 5 June 2021) were available.

**Table 1.** Details of seamount sampled and abundance of *Diastobranchus capensis* in photographic imagery.

| Seamount | Seamount Peak Depth (m) | Depth Range Sampled (m) | SURVEY #1 | | SURVEY #2 | | SURVEY #3 | |
|---|---|---|---|---|---|---|---|---|
| | | | No. Images | Average No. *D. capensis* per m$^2$ | No. Video Transects | Average No. *D. capensis* per h | No. Video Transects | Average no. *D. capensis* per h |
| Patience | 817 | 904–1240 | 123 | 0.259 | 2 | 220.0 | 2 | 69.59 |
| Corvina Group | 750 | 750–1250 | 0 | – | 0 | – | 6 | 2.07 |
| Z5 | 1178 | 1176–1250 | 22 | 0.000 | 0 | – | 1 | 0 |
| Pedra | 714 | 725–1343 | 448 | 0.002 | 0 | – | 7 | 4.19 |
| Mongrel | 723 | 778–1244 | 286 | 0.001 | 0 | – | 0 | – |
| Z16 | 1004 | 1057–1404 | 54 | 0.000 | 0 | – | 14 | 1.15 |
| Sisters | 807 | 864–1350 | 458 | 0.001 | 0 | – | 10 | 6.7 |
| Hill B1 | 1064 | 1139–1405 | 60 | 0.003 | 0 | – | 0 | – |
| Hill K1 | 1238 | 1248–1817 | 251 | 0.001 | 0 | – | 8 | 7.82 |
| Hill U | 1083 | 1099–1413 | 247 | 0.002 | 0 | – | 11 | 6.29 |
| Dory Hill | 1054 | 1091–1469 | 113 | 0.000 | 0 | – | 0 | – |

*2.3. Estimating Fish Abundance*

We scored the abundance of fishes in two ways:

(1) Density: counts of individual fishes above (overlapping) polygons of known area were made during Survey #1 [35] for *D. capensis* and other fishes: predominantly orange roughy (*Hoplostethus atlanticus*), combined Oreosomatidae (*Allocyttus verucosus* and *Neocyttus rhomboidalis*); squaliform sharks, and macrourids. Polygon areas were estimated by using the stereo configuration of the video to determine absolute areas of seafloor in the high-resolution still images. Surface areas in the 3-dimensional perspective of the oblique field of view were calculated from calibrated stereo imagery using Photomeasure software (SeaGIS-TM V3.15 & SeaGIS-EM V5.51, www.seagis.com.au/index.html, accessed on 5 June 2021). Polygons of known area were drawn in the images by identifying a series of common points in corresponding frames from the stereo video and the still images. These polygons encompassed an average area of 5.0 m$^2$. Differences in mean density between seamounts used a 2-tailed *t*-test.

(2) Relative abundance: counts of individual *D. capensis* in the field of view in non-overlapping video-frames (5 s intervals) to provide an along-transect estimate of the total number of individuals. This provided a conservative sum of individuals because eels were not counted in the gaps between images. For survey #3, the video was viewed at half-speed and all eels in the field of view were counted. The observed number of eels was standardised to individuals per h of video tow.

*2.4. Reproductive Biology and Swimbladder of Diastobranchus capensis*

Most eels were dead when retrieved but a few moribund individuals were euthanized in a 100 mg/L Aqui-S solution until non-responsive. They were then weighed to the nearest gram and measured (total length, TL) to the nearest millimeter and dissected to examine (1) the size and structure of the swimbladder to relate to its acoustic reflectivity, and (2) the gonads to determine sex and numerically score the reproductive status. Gonads were removed and weighed to the nearest gram to calculate the GSI = gonad weight/total (non-eviscerated) fish weight. Ovaries were fixed in 10% formalin for fecundity estimates. Fecundity estimation was as follows: each ovary was drained through 300-micron nylon mesh and rinsed with freshwater, drained again and gently patted with absorbent towel to

remove excess liquid. Preserved ovaries were weighed to nearest 0.1 g and subsampled (2 g tissue) in the anterior, posterior and middle sections. Subsamples were rinsed again in freshwater through 250-micron nylon mesh to remove as much formalin as possible and weighed to the nearest 0.1 g. Each subsample was placed separately into a circular (plankton) counting tray and individual eggs were teased away from integumental tissue and counted; the diameter of 10 oocytes from each subsample was measured using an ocular micrometer on a dissecting microscope to an accuracy of 0.1 mm. The data were scaled up to total egg count per ovary (=estimated fecundity) after confirming that counts from the three sections of the ovaries were from the same statistical distribution using the non-parametric Kruskal-Wallis test. Eel bodies were examined for evidence of changes to dentition and skin degradation associated with spawning. A linear regression was used to test for correlation between GSI and fish length.

## 3. Results

### 3.1. Abundance of Diastobranchus capensis

Numerous *D. capensis* were observed in the camera tows on Patience Seamount during each of the three surveys in 2007, 2015 and 2018 (surveys #1 to #3, respectively). More than 25 individual eels were observed in individual images during surveys #1 and #2 (e.g., Figure 2a), and as many as 13 *D. capensis* individuals counted over a single image polygon during survey #1. The relative abundance of *D. capensis* along the three camera-transects made in surveys #1 and 2 (March/April) (Figure 3) was very high: standardised counts (eels per h of camera tow) were 278, 362, 162 (mean = 267); counts were lower (17, 123; mean = 70) along two transects made during survey #3 (December) (Figure 3). The total (unstandardised) counts of eels per transect were 763, 931, 475, 59, 167 (order as above); the counts at 5-sec intervals are mapped along the transects to provide spatial context (Figure 3). Within the combined depth range surveyed (755 to 1432 m) across all three surveys at Patience Seamount, the complete distributional depth range of *D. capensis* was ~900 to 1250 m, but the great majority (>95%) were observed in ~1000 to 1200 m depths (Figure 4). Both our measures of abundance are conservative because the still images and video frames in which eels were counted do not overlap, i.e., there are gaps between them.

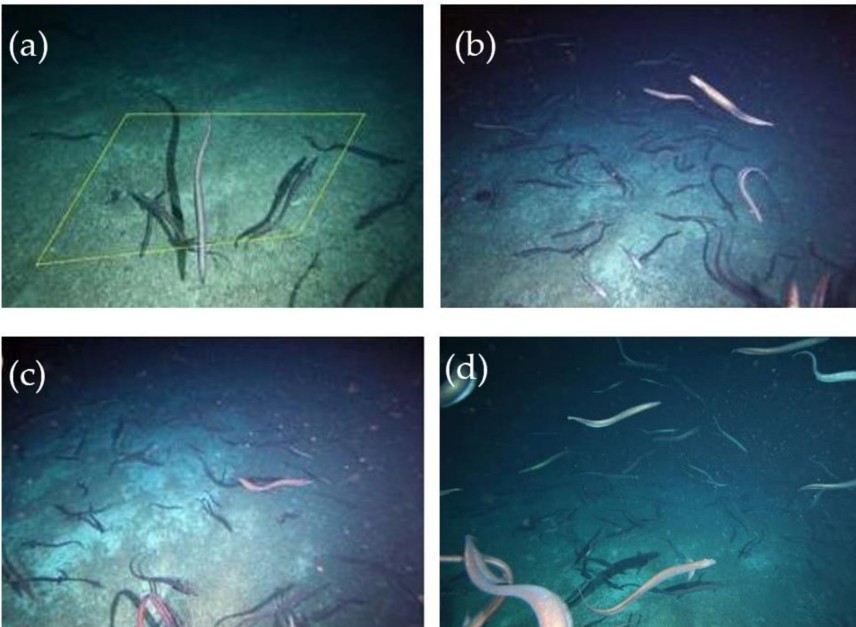

**Figure 2.** Near-seabed images from Patience Seamount in ~1100 m depth (camera transect 054 on Survey #1): (**a**) showing polygon method for calculating fish density (here, four eels overlap the vertical extension of the polygon; (**b**–**d**) showing the fish assemblage is highly dominated by *Diastobranchus capensis*.

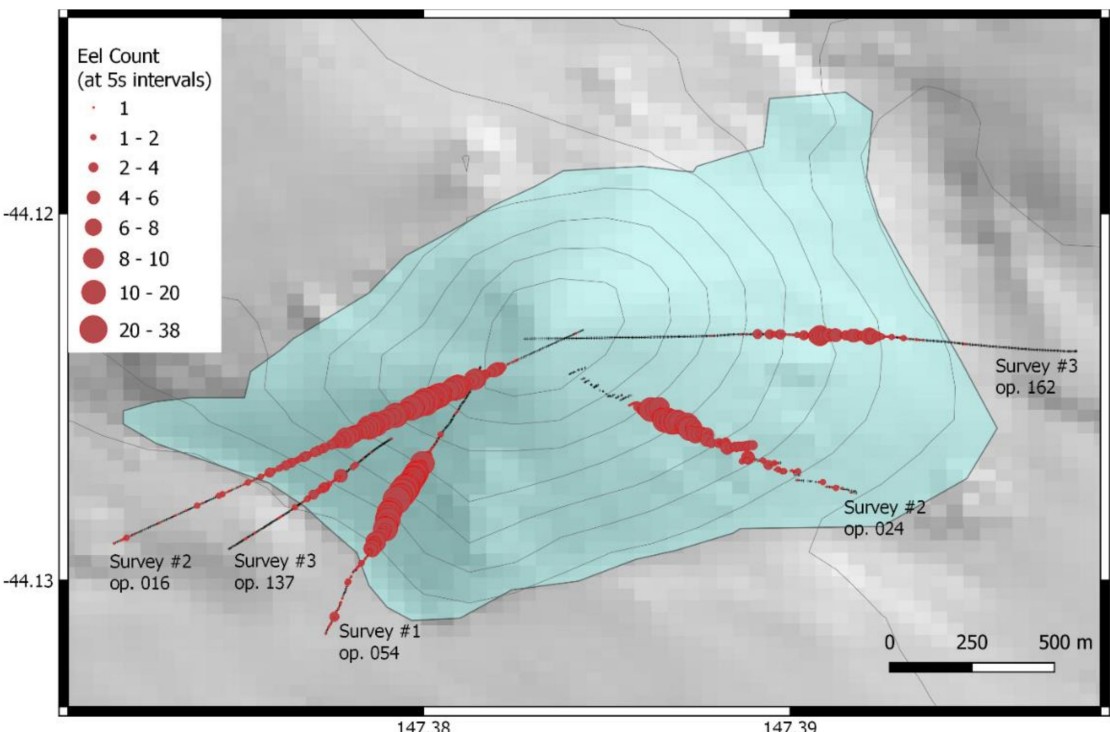

**Figure 3.** Map of the distribution of counts of individual *D. capensis* on Patience seamount in the field-of-view of the video camera at 5 s intervals; mean raw counts from the five transects (anti-clockwise from left) = 763, 59, 931, 475, 167; zero counts are shown as black crosses. Seamount base (blue shading) is defined by rate of change of slope; depth contours = 50 m intervals from 850 m.

The numbers of *D. capensis* observed on Patience Seamount were relatively very high compared to 10 other adjacent seamounts (Figure 1b). When measured during survey #2, the mean *D. capensis* density was 0.26 individuals $m^{-2}$ (+/− 0.038 SE) (Figure 5a) and statistically highly significantly different between seamounts (F = 41.369, df 10,1740, $p < 0.001$). The relative abundance of *D. capensis* was also relatively high during survey #3 (Figure 5b), and similarly to surveys #1 and #2, only lone *D. capensis* were seen on the other seamounts. The density of *D. capensis* on Patience Seamount during survey #2 was more than double the density of all other fishes combined on that seamount and any of the other nine adjacent seamounts surveyed (Figure 5a).

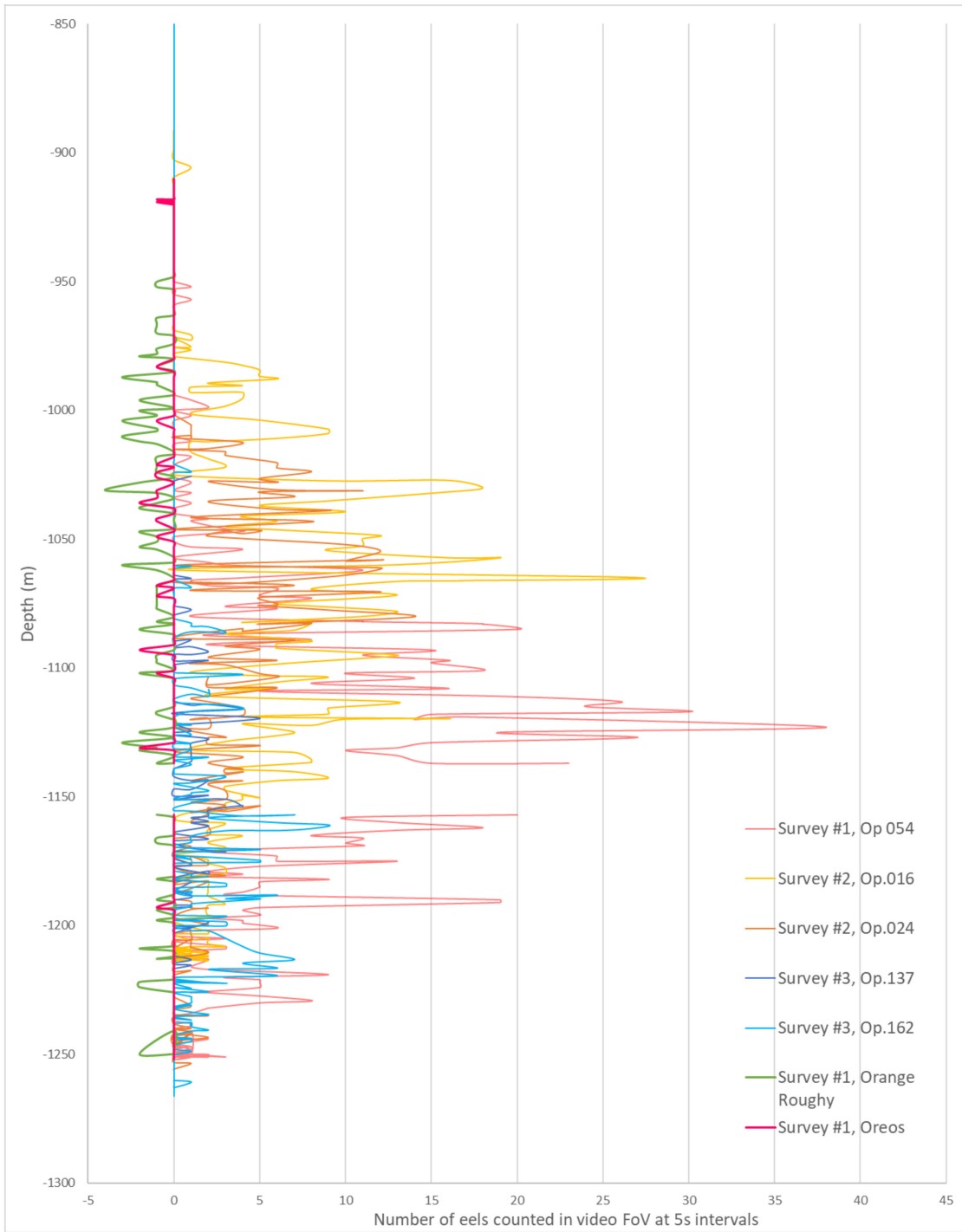

**Figure 4.** Depth distribution of dominant fishes on Patience Seamount based on counts of individuals along video transects: right-hand side of plot—*D. capensis* from Surveys #1 (March 2007), #2 (April 2015), Survey #3 (December 2018) (see Figure 3); left-hand side of plot—orange roughy and oreo dories from Survey #1.

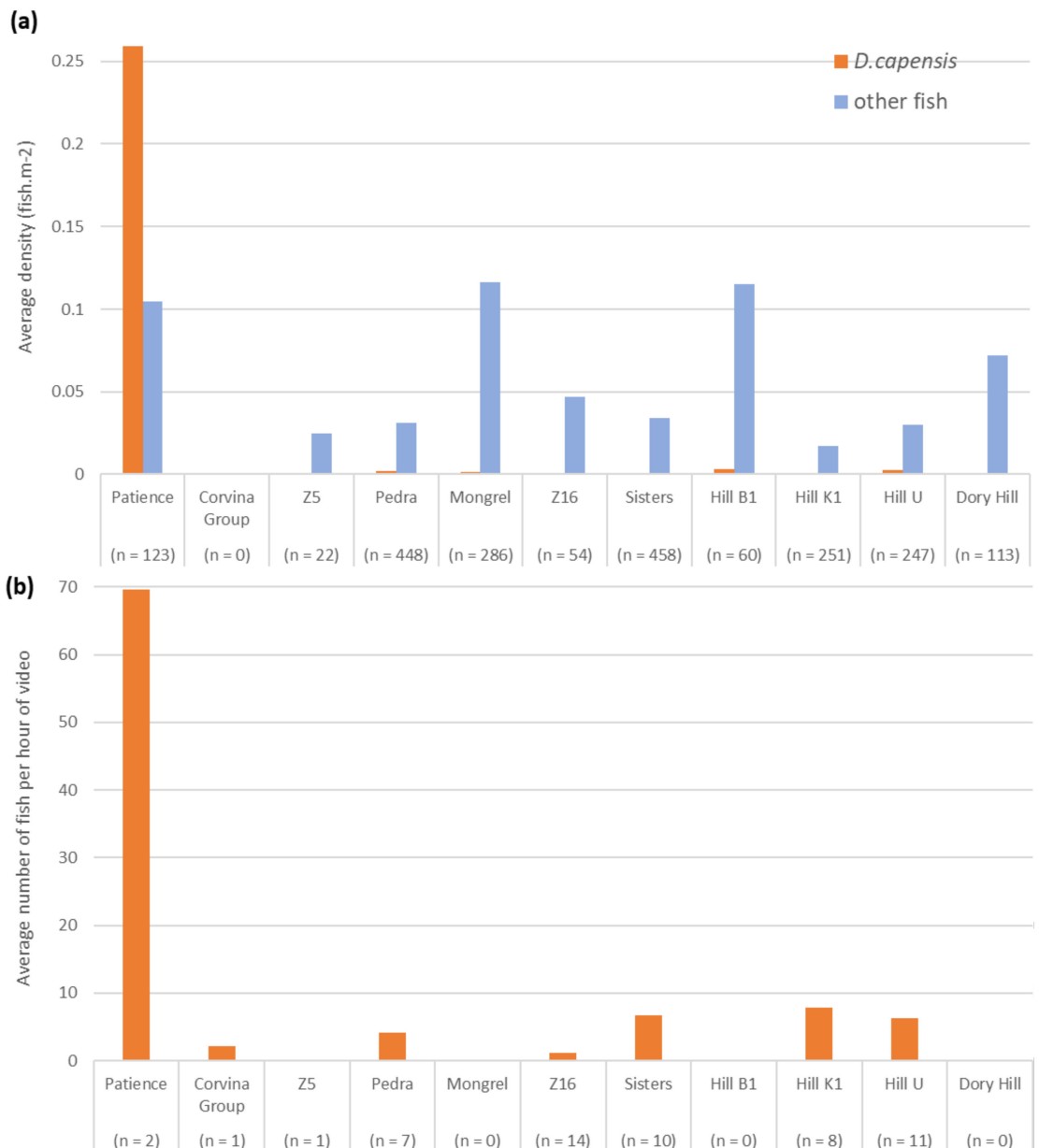

**Figure 5.** Observations of *D. capensis* on seamounts south of Tasmania (see Figure 1). (**a**) Density counts of *D. capensis* and other fishes derived from still images with polygons of measured area collected during survey #1, densities for each seamount are averaged over the number of images (n) examined. (**b**) the number of *D. capensis* observed per h of video taken during survey #3, densities for each seamount are averaged over the number of video transects (n) examined.

### 3.2. Reproductive and Swimbladder Characteristics of Diastobranchus capensis

A total of 20 adult *D. capensis* specimens were captured in three beam trawl samples during survey #2 in April 2015 (18 in a single sample at 1087 m depth within the Patience Seamount aggregation, and two on the adjacent slope at 1070 m and 1060 m depth) comprised 14 females (782–1204 mm TL and 632–2296 g weight) and six males (653–780 mm TL and 338–588 g). Two other adult females were caught from Patience Seamount by hook and line during survey #3 in December 2018 at ~1019 m. All 20 individuals from 2015 had gonads in an advanced stage of spawning condition (Figure 6) and empty stomachs; in contrast, both females caught in 2018 were in a non-spawning condition. The gonads of gravid females and males had a multi-lobed structure and were very large relative to body size—extending along the entire length of the body cavity (Figure 6). The ovary walls were

transparent, and oocytes were clearly seen with the naked eye, but the oocytes were not free and did not flow freely from the cloaca. The testes were firm and white; milt flowed freely from the cloaca when gentle pressure was applied to the body cavity. Fresh ovary weight ranged from 149–537 g, and testes from 28–138 g. The GSI ranged between 17.4 and 25.8 in females, and 6.8 to 24.8 in males; there was no significant correlation between GSI and eel length ($R^2$ = 0.10). Individual fecundity in the mature females, estimated from egg counts in subsamples taken from the anterior, mid- and posterior regions of ovaries from each of five individuals that represented the overall eel size range, ranged from 73,339 to 235,061, with the number of eggs increasing linearly with increasing fish length ($R^2$ = 0.92). The average egg diameter (n = 10) was consistent within and between gonads at 0.9 to 1.0 mm. The mean (n = 15) was 469 eggs per gram (range = 409 to 538; SE = 10.47). There was no significant difference between location of the tissue collection within the ovaries, despite some considerable variation in the counts (Kruskal-Wallis *p* = 0.65).

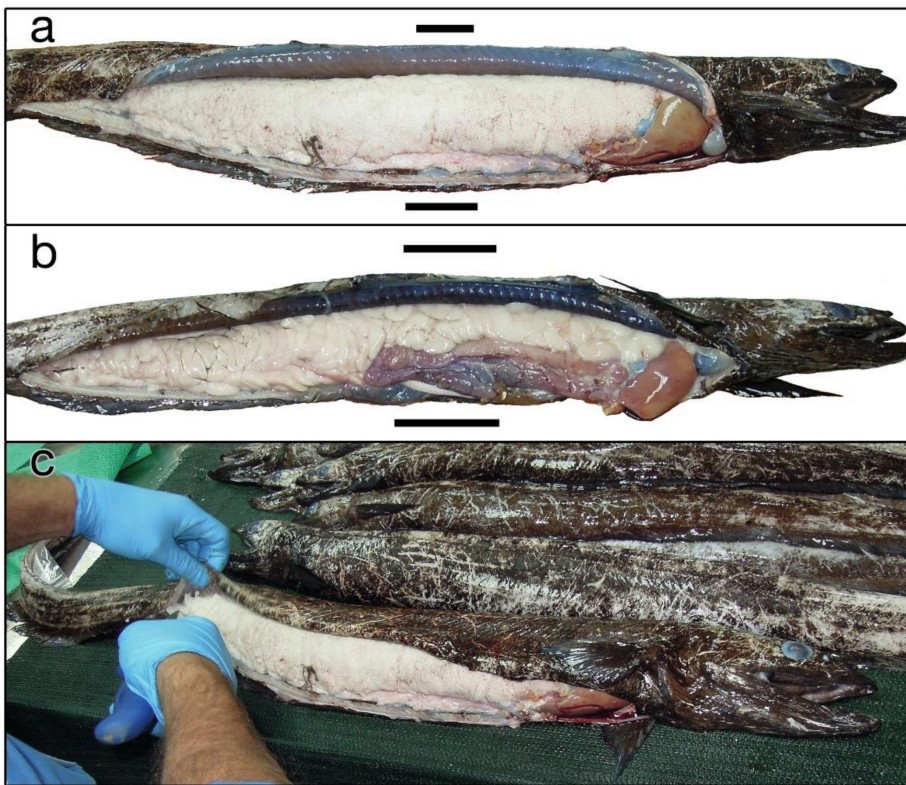

**Figure 6.** Partially dissected *Diastobranchus capensis* specimens from Patience Seamount showing (**a**) ovaries in abdominal cavity of female; (**b**) testes in abdominal cavity; (**c**) several other individuals with external damage consistent with net capture. Scale bars 50 mm (account for near/far distortion).

All eels observed (spawners, and non-spawners in and away from the aggregation) were generally slow moving and did not show distinct flight response away from the camera unless they came very close to it. We observed no particular indications of spawning behaviour (such as tight grouping or egg shedding) by spawning eels, and there were no external signs of spawning condition such as skin degradation or changes to dentition; scraping and other damage to skin was consistent with net capture (Figure 6).

Dissection of *D. capensis* revealed it possesses an exceptionally long gas-filled swimbladder (approximately 2/3 of its body length) (Figure 7). The swimbladder has a relatively wide anterior bulb that tapers over the length of the posterior tube.

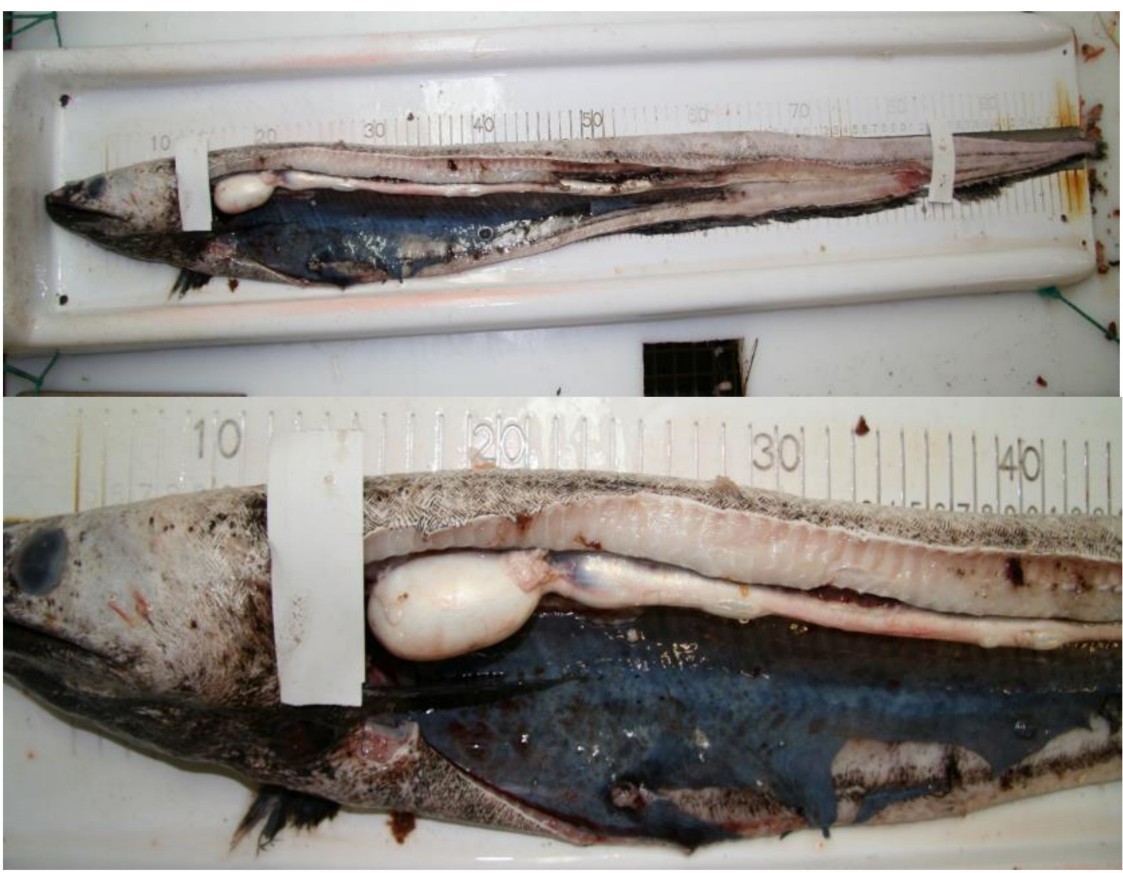

**Figure 7.** Partial dissection of *Diastobranchus capensis* to reveal the shape and extent of the gas-filled swimbladder. In this individual of 102 cm total length, the swimbladder is 67 cm long (shown by inner edges of the two white markers); it is widest (4 cm) at the anterior bulb then tapers over the length of the posterior tube.

### 3.3. Characteristics and Composition of Acoustic Marks at Patience Seamount

During survey #2, echo-soundings at both 38 and 18 kHz identified strong, extensive, and persistent 'acoustic marks' encircling the flanks and base of the Patience Seamount, consistent with the presence of a feature-associated fish aggregation 1000 to 1250 m depths. Echo-soundings made down the south-east side of the seamount during video tow 024, Survey #2 on April 12 (Figure 8a) showed only low signal backscatter of biological origin instead of expected high backscatter consistent with the high numbers of *D. capensis* observed during this tow; this was attributed to the mismatch between the camera observation < 10 m above seafloor and a high acoustic dead-zone ~100 m above seafloor (Figure 8a). A much larger and stronger aggregation was observed later on 12 April on the north-west flank of the seamount (when raw acoustic data were recorded successfully) (Figure 8b). This aggregation was within the expected depth range of *D. capensis*, and approximately 100 m high long 1000 m, extending away from the seafloor and above the dead-zone region. Images from the star-pattern transects showed the aggregation consistently associated with the Patience Seamount in the depth range of approximately 1000 to 1250 m (Figure 9).

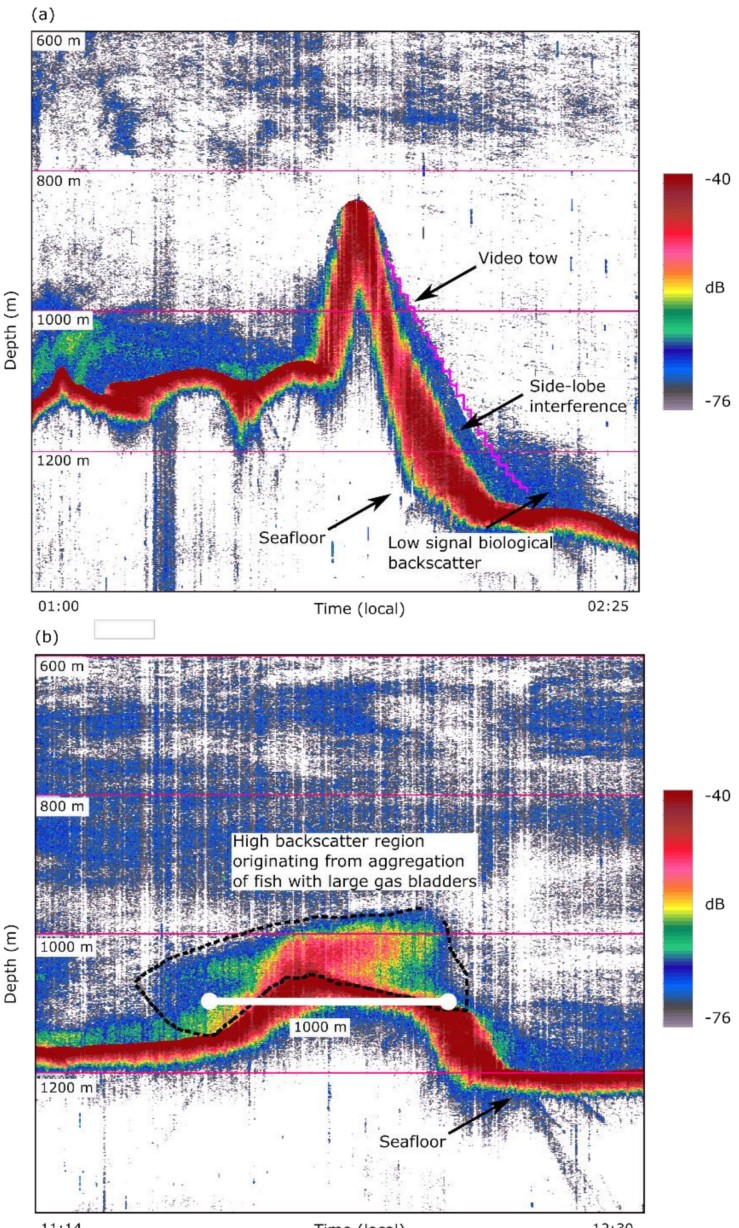

**Figure 8.** Echosounder (18 kHz) volume backscatter echograms from Patience Seamount showing: (**a**) echogram recorded on 10 April 2015 commencing at 01:00 (local time) during camera transect 024 (Survey #2) down the south-east flank (magenta line shows the approximate location of the video tow); a region of low backscatter of biological origin can be observed at the end of the transect, the acoustic dead-zone and side-lobe interference can mask signal of biological origin; (**b**) echogram recorded on 10 April 2015 commencing at 11:15 (local) showing a region of very high backscatter that originates from fish with high target strength diagnostic of large gas-filled swim-bladders; dimensions of the aggregation (~100 m high × ~1000 m long) are indicated by the vertical axis and the horizontal scale bar. Right-hand side colour bar indicates decibel values associated with echogram colours.

Camera observations confirmed that *D. capensis* was highly dominant (Figure 2) over the total depth range surveyed (755–1432 m), with the great majority of individuals seen in 1050 to 1200 m depths on the flank and base of Patience Seamount corresponding to the depth of the acoustic marks (Figure 4). Other fishes were photographed on Patience Seamount, and collectively made up 29% of total observations (Figure 5). However, only two taxa occurred at relatively high abundance: orange roughy and oreo dory (mostly

juvenile *Allocyttus verrucosus*), both of which were scattered at low density in a similar depth range to *D. capensis* (Figure 4). A beam trawl tow targeted at the aggregation in 1087 m depths caught 20 *D. capensis* but no other fish species.

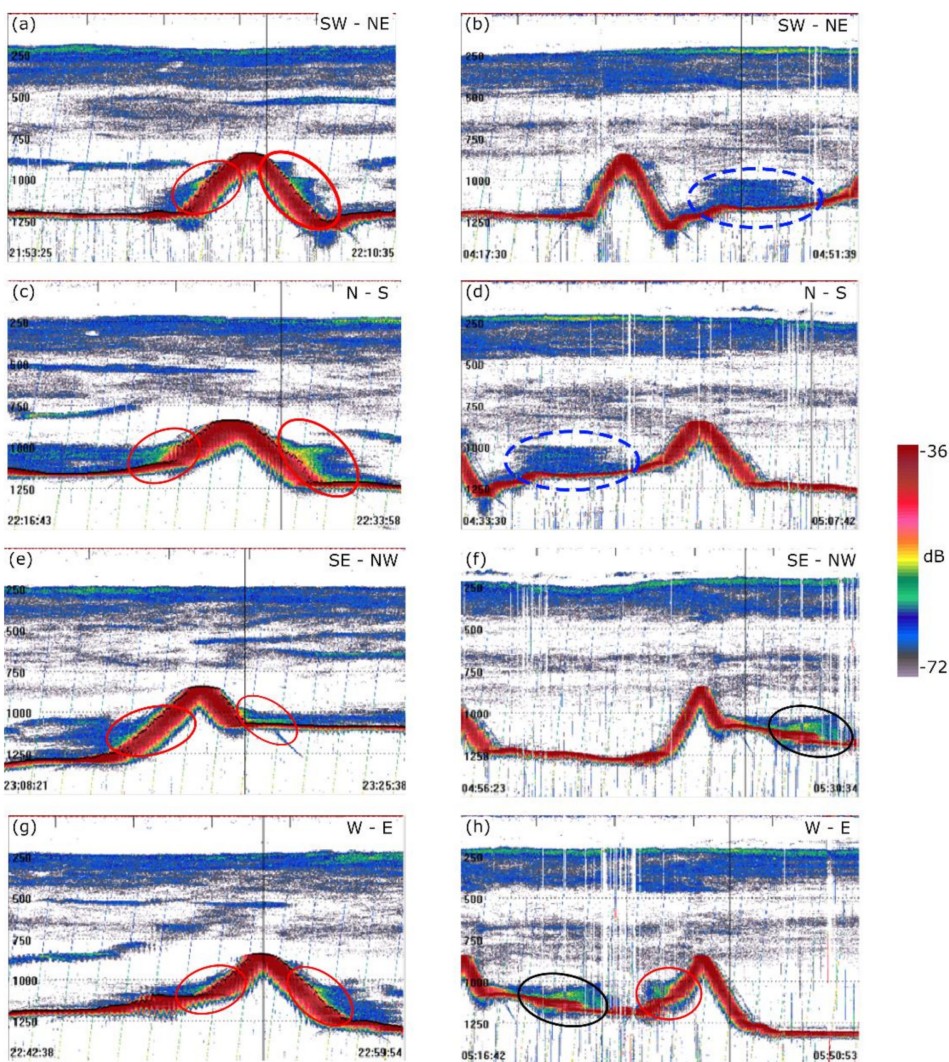

**Figure 9.** High signal acoustic backscatter regions believed to be *Diastobranchs capensis* (red circles, panels **a**, **c**, **e**, **g** and **h**) in the two 'star-pattern' surveys. Other low signal backscatter regions, e.g., panels (**b**) and (**d**) (blue dashed circles) are unlikely to be *D. capensis*. High signal regions away from the seamount feature itself also believed to be *D. capensis* due to the absence of other high signal candidate species are marked separately (black circles). Images from real-time display taken from Simrad ER60 software running the Simrad EK60 38 kHz echosounder. Display range is estimated to be −36 to −72 dB.

## 4. Discussion

### 4.1. Characteristics of the Aggregation of Diastobranchus capensis

Multiple lines of evidence substantiate our conclusion that there is a large aggregation of the basketwork eel, *D. capensis*, on Patience Seamount. Its existence was first detected by commercial fishers in the early 1990's and was present, at varying magnitudes, during our three visits in 2007, 2015 and 2018—a span of some 25 years. Thus, it appears to be persistent through time, present year-round (based on observations in March, April and December), and much larger in the austral autumn (around April). All 20 eels captured during the 2015 austral autumn voyage were in an advanced stage of spawning, with some 'running ripe' indicating readiness for imminent spawning, and strongly indicating the

large aggregation seen at that time was for spawning purposes. In addition, all 20 had empty stomachs which indicated this was not a feeding aggregation, whereas the two eels caught from the smaller aggregation in December 2018 using baited long line were both feeding and in non-spawning condition.

The abundance of the eel in the aggregation was very high as measured at seamount, local and regional scales. Thus, the mean *D. capensis* density (0.26 individuals per m$^2$) in the spawning aggregation (April 2015) was 40 to 185 times greater than its density at the other adjacent seamounts where it was present, and 2 to 15 times greater than the combined density of all fish species at any other adjacent seamount (Figure 5). Our examination of research and commercial catch records validated by sea-going observers for *D. capensis* throughout its range (collectively ~2500 trawl operations from North Atlantic, Chile, New Caledonia, New Zealand and southeastern Australia [21,36,37] showed most recorded catch rates were very low (>90% were <20 kg/nm, or 0.5 gm$^{-2}$) but that that there were two exceptional catches. These were 286 kg/nm (7.72 gm$^{-2}$) from the continental slope immediately adjacent to the Patience Seamount, and the largest of 476 kg/nm (12.85 gm$^{-2}$) taken in September 2000 on the 'Smith City' Seamount on the Chatham Rise off New Zealand [data in 21]. The species' wide range (Antarctic Convergence to Arctic) and frequent occurrence in its core distribution, e.g., 85 and 90% of all trawls examined, respectively off New Zealand and Australia [21], signals there are undoubtably other spawning sites. Clues to their general locations are given by museum records (author's data) of two gravid females, one each from the Great Australian Bight and southwest Indian Ocean, and possibly by the large catch on Smith City Seamount—although no spawning fish were reported.

We are confident hydro-acoustics have provided a way of scaling the size and dynamics of the aggregation at Patience Seamount. Often caution is required when attributing acoustic marks detected by echosounder to particular species because a mix of species (fishes and invertebrates) with varying acoustic target strengths may be present [38]. Further, ground-truth sampling with cameras can also be problematic because individuals filmed close to the seabed may be in the acoustic 'dead-zone' and remain undetected by echo-sounding [39]. This 'dead-zone' is generated by the off-axis return acoustic backscatter signal from the seafloor arriving at the echosounder receiver earlier than biological signal from nadir and is further increased by 'side-lobing' on the steep flank of seamounts. Dead-zones may extend to as much as 100 m from the seabed when using a hull-mounted echosounder in steep deep-sea environments but can be reduced using acoustic systems deployed on towed platforms that reduce the range between echosounder and seafloor [40]. In our data, large numbers of *D. capensis* seen in camera transect 024, Survey #2 did not generate an intense acoustic mark as expected (Figure 8a), clearly indicating that the eels were in the acoustic dead-zone. Nonetheless, the strong marks seen around the Patience Seamount at various other times (e.g., Figure 8b) can be confidently predicted to be *D. capensis* due to (1) its expected very high echo intensity (target strength) relative to other co-occurring species, and (2) the lack of any other plausible target.

Attributing high echo intensity (TS) and the high backscatter regions seen at Patience Seamount to *Diastobranchus capensis* is corroborated by the morphology and size of its swimbladder. Fish TS is strongly correlated with the gas-water density contrast of the swimbladder and its cross-sectional area [41], with large and long gas-filled bladders producing high acoustic backscatter [42]. These two features are accentuated in *D. capensis* which is large-bodied and elongate (~1 m in length) and possesses an exceptionally long gas-filled swimbladder (approximately 2/3 of its body length) (Figure 7). Model estimates of TS for a 1 m long *D. capensis* using parameters provided in [38] give a tilt-averaged TS of −30.1 dB. This is a very high value relative to the TS of co-occurring species: oreos with a TS—one order of magnitude lower [38], and orange roughy with a TS—two orders of magnitude lower [43]. The high backscatter regions observed at Patience Seamount can only have been generated by either a species with an exceptionally high target strength, or by another species with lower target strength but relatively very high density. The latter

is implausible because the only other relatively abundant and schooling benthopelagic fishes that co-occur on Patience and adjacent seamounts are orange roughy and, to a much lesser extent, the shallow oreo species (*A. verrucosus* and *N. rhomboidalis*). For these species to produce the high backscatter observed, their numeric densities would need to be numerically higher by factors of ~10 and ~100, respectively. There is no evidence of high densities of these species in camera tows, and no expectation of observing large aggregations of orange roughy outside their austral winter spawning period in June/July.

While it is not yet possible to precisely quantify the population size within the *D. capensis* aggregation at Patience Seamount, our observations show that (1) eel density is very high in areas around the main aggregation, with conservative counts of hundreds of eels along narrow camera transects (field-of-view ~5 × 5 m) of ~1 to 2 km in length, and (2) the main aggregation is of the order of 100 m high and >1 km in length (and unknown width) in echograms (Figure 8b). It is therefore both reasonable and conservative to estimate that tens of thousands of *D. capensis* comprised the spawning aggregation at the point in time when it was surveyed in April 2015. The total spawning population visiting the Patience Seamount may be much greater if there is an extended period of spawning and turn-over is accounted for. In any event, for *D. capensis*, a species that is widespread and common at typically very low density, the magnitude of this single substantiated aggregation strongly suggests high spawning-site fidelity and regional-scale catchment for spawning. This is substantiated off southeastern Australia where there is no other record of an eel aggregation from a decades-long documented commercial catch history, anecdotal reports from fishers, or many scientific surveys. Despite the duration of spawning, turn-over dynamics and fluctuations in year-round eel abundance all remaining unknown, our observations suggest that *D. capensis* migrate from large distances along the narrow mid-continental slope (e.g., several 100 s km around Tasmania alone) to the small Patience Seamount (base area of 3 km$^2$) to spawn.

Although we have not attempted to quantitatively estimate total eel numbers in the aggregation using acoustics due to an unvalidated estimate of *D. capensis* TS (echo intensity) and dead-zone effects, it appears to be the largest observed natural aggregation of a deep-sea eel. Notwithstanding, any estimate would not be directly comparable to other reports of high eel density from bathyal and greater depths, e.g., [15], because none are of a spawning aggregation, and because most are made with baited cameras that attract fish then extrapolate fish density over broad areas using a variety of different metrics and methods (see review by [15]).

### 4.2. Ecological Status of the Diastobranchus capensis Aggregation

This population of *D. capensis* was exposed to impacts from a seamount-focused fishery around Tasmania over a period of some 30 years, starting around 1979. The species was caught as bycatch during the peak of the fishery between about 1985 and 2006 when some 47,000 bottom trawling operations reduced the target stock of orange roughy to ~10% of pre-fishing biomass [44,45], and over 600 trawls were made on the Patience Seamount [32]. Anecdotal (unrecorded) reports of eel bycatch on Patience Seamount, much of it from trawls targeted at acoustic marks mistaken as orange roughy, included individual catches of 'many tonnes' (i.e., many 1000's of individual eels), and were the origin of the fishing industry moniker 'Eel Hill' (AW, unpublished data). Bottom trawling also had widespread impacts on seamount habitat, including at Patience Seamount where disintegration and removal of stony coral reefs resulted [32]. Total fishing mortality on *D. capensis* is unknown but was probably substantial; any estimate based on recorded bycatch would be a great underestimate.

Although the impacts of past fishing upon the eel population associated with Patience Seamount cannot be substantiated due to a lack of baseline data, it is likely to have had a negative effect. High vulnerability of many deep-sea species to exploitation is suggested to stem from a tendency to aggregate on seamounts, and by possessing life-history traits characteristics such as large body size, slow growth, late sexual maturation and low



natural mortality that make them vulnerable to exploitation [46,47]. However, there is a continuum in these life-history characteristics among deep-sea fishes [48] and *D. capensis* appears to be 'mid-range' in many respects. Thus, while it is large-bodied (up to 1.8 m, and the largest synaphobranchid eel [17] and has low natural mortality (0.144 [49]), it has a maximum recorded age of 47 years and was not considered long-lived [49], and appears to be sexually mature by about 10 years old (our gravid individuals compared to the age-length curve of [49]). Nonetheless, significant declines of *D. capensis* have been documented in New Zealand waters where it is also an incidental catch in orange roughy and oreo fisheries, strongly implying that fishing can have a deleterious effect on this species [50]. In the North Altantic, significant fishing-related declines of the related basketwork eel *Synaphobranchus kaupi* were reported [51,52] and benefits to scavenging synaphobranchid eels from fishery discards were offset by mortality from bycatch [52]. The significance of such fishing-induced impacts may have implications both for *D. capensis*, and for surrounding ecosystems. Spawning location is the anchor of a fish's spatial distribution and will affect the degree to which a species can respond to environmental variability by changing habitats [53]. This would indicate that high fidelity to a single small spawning seamount by a regional-scale population will confer a low ability to respond to varying environmental conditions whether caused naturally or human-induced. It is likely that *D. capensis* is an important component of deep-sea ecosystems given its role a carrion eater [27], and by providing a substantial supply of detrital material to the benthic community [54].

Since this exposure to fishing mortality and habitat alteration, Patience Seamount became part of the 'Huon' Australian Marine Park (AMP) in 2007 and is now off-limits to trawling. Our surveys in 2015 and 2018 (made after Patience Seamount was protected) show that the spawning aggregation of *D. capensis* was still large. We, therefore, hypothesise that the southeast Australian population of *D. capensis* is stabilising and in a phase of recovery.

*4.3. Conservation Value and Recovery Monitoring*

Australia is a global front-runner in marine conservation having established a large network of offshore Australian Marine Parks (AMP) since 2007 [31]. Network design recognised the significance of seamount biodiversity and is the primary conservation value of the Huon AMP that encloses most of the cluster of small seamounts containing Patience Seamount. Monitoring conservation values is a priority activity for managing the AMP network, and a national strategy of Monitoring, Evaluation, Reporting and Improvement (MERI) will be implemented soon [55]. The aggregation of *D. capensis*, undocumented when the Huon AMP was designed and declared, demonstrates the importance of small spatial-scale features and phenomena to this process. The aggregation is an important and tractable indicator for monitoring a biological response to protection because it is fixed in space, has a known temporal signal, and can be quantitatively and cost-effectively measured by acoustics with little further extractive sampling. The objectives for monitoring should include to fully validate its status as a spawning aggregation and reveal presently unknown aspects of its ecology. It is also an important performance indicator of management intervention because fish spawning aggregation sites are being impacted worldwide due to fishing [56]. Monitoring the status of the basketwork eel spawning aggregation at Patience Seamount provides an unprecedented opportunity to understand a conservation-led recovery in the deep sea.

**Author Contributions:** Conceptualization, A.W.; Data curation, J.P.; Formal analysis, D.O.; F.A. and T.R.; Methodology, A.W., T.R. and M.G.; Project administration, A.W.; Writing—original draft, A.W. and D.O.; Writing—review and editing, A.W., D.O., F.A., T.R., M.G. and J.P. All authors have read and agreed to the published version of the manuscript.

**Funding:** This research was funded primarily by CSIRO Oceans and Atmosphere. Survey #3 was funded, collectively, by the CSIRO Oceans and Atmosphere, Parks Australia and the Australian Government National Environmental Science Program (NESP), Marine Biodiversity Hub.

**Institutional Review Board Statement:** The study, including biological collecting, was conducted in accordance with the following permits: Animal Ethics Permit (2018-26); Commonwealth Waters Permit; Australian Marine Parks Permit; and Australian Fisheries Management Authority Permit.

**Informed Consent Statement:** Not applicable.

**Data Availability Statement:** Metadata for the data collected for this study is available through Marlin the CSIRO Oceans and Atmosphere metadata tool http://www.marlin.csiro.au/, accessed on 5 June 2021; for access to imagery and annotation data please contact the corresponding author.

**Acknowledgments:** We acknowledge the committees and staff of Australia's Marine National Facility (MNF) for access to the research vessels RV Southern Surveyor and Investigator, and the captain and the crew of the vessel for their hard work in making the survey successful. We also thank the sea-going staff from the MNF and many colleagues from CSIRO Oceans and Atmosphere and Parks Australia whose support was vital to the field collection of data.

**Conflicts of Interest:** The authors declare no conflict of interest.

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
