# Peer review of "A Very Large Spawning Aggregation of a Deep-Sea Eel: Magnitude and Status"

_jmse, doi:10.3390/jmse9070723_

Round 1
Reviewer 1 Report
This manuscript reports observations of an aggregation of spawning eels (Disatobranchus capensis) on the Patience Seamount off Southeast Australia. The authors used video imageries and fishery stocks data collected during 3 field surveys in 2007, 2015 and 2018, and echosoundings acoustic data that corroborate the detection of eel aggregation in the area. The authors also dissect eels to examine the internal anatomy that further support the reproductive characteristics of the captured individuals. Despite the shortfall that the study was unable to provide a quantitative estimate of the population size, the data appear to support the postulation of an eel spawning aggregation. This work is of the interest for JMSE’s readership, and I recommend publication with minor revision.
My major comment resides in the Discussion section. The authors start with a caution on their interpretation of the data and conclusions drawn from limited spatial coverage and uncertainties associated with technical challenges in acoustic signals. This is appreciated and helpful to future research efforts. The paper ends with a trite discussion on the impact of overfishing and a note of conservation (sect 4.2 and 4.3) which, in my opinion, weakens the value of this work. It would be a lot more interesting for the authors to discuss the pros and cons of their sampling design and strategies. Why 3 separate surveys that spans over 11 years (2007-2018), why those 3 years in particular? Why was the study not designed to optimize the detection or quantification of the population size? Is the detection of this small eel spawning ground a result of a carefully executed experiment, or a coincidence? What types of sampling strategy do the authors recommend in order to uncover other ‘undoubtably’ spawning sites? What the authors must do to speak beyond reasonable doubt, and to not reply on speculative languages such as ‘seems likely’ or ‘seems reasonable to hypothesize’? There remain many unanswered questions that I wish to see the authors to comment on in order to move the science forward, rather than falling back on the truism of the impact by overfishing.
Author Response
We are very pleased to have received three thoughtful reviews that each recommended only minor changes. These have enabled us to make some beneficial changes to our manuscript. The key issue for this paper, identified by all reviewers, is whether or not this aggregation can be confidently identified as a spawning aggregation. Each reviewer considered that our multiple-lines-of-evidence approach to interpreting the data was sound, that there was strong evidence for a spawning aggregation, but that it wasn’t conclusive.
To provide a conclusive account that fully characterised this aggregation and spawning activity would require multiple seasonal surveys using quantitative acoustics and capture of eels for full histological analysis. That kind of research effort can only be applied to species supporting industrial-scale fisheries (e.g. orange roughy), or to the highly valuable, enigmatic and elusive anguillid eels. We feel that our data set supports our conclusion, and that our manuscript finds the right balance between showing confidence in the data, and tempering this with a degree of caution. We also note that our approach is rather conservative alongside other studies of eel spawning aggregations cited in our Introduction that are based on considerably fewer data. Notwithstanding, we have made some adjustments to the text in the Abstract, Introduction and Discussion to emphasise that future validation of our conclusion is required. We have done this without watering-down the conclusion and have used less-speculative language as advised by the reviewers. Against this justification, we have not withdrawn the conclusion from the Abstract or Title.
More specifically, we do not agree with the assessment of Sections 4.2 and 4.3 as trite for the following reasons. This deep-sea eel aggregation is extraordinary in being the first described in world oceans. We were able to provide detail, including quantitative data, about the extent of fishing impact on the aggregation, something that is rarely possible especially in a deep-sea setting. This provides context to the unique history of this aggregation — heavily impacted but now protected — and identifies a very worthy target of future quantitative monitoring to answer important questions about species and systems resilience and recovery in the deep sea. Part of our aim in providing this information in the Discussion is to stimulate interest in this type of monitoring — the only way that addition information could be collected to validate our conclusions about spawning.
The reviewer lists some very interesting points about survey design – but unfortunately missed our disclosure early in the methods that our data were collected opportunistically during surveys that had broader objectives. We have now emphasised this point in the Introduction.
As noted above, we provide a measured and conservative interpretation of an extensive collection of data. We believe our work substantially moves forward a range of science issues, and provides the basis — and stimulus — for future work to address the key issues: quantitative assessment of eel abundance using remote calibrated hydro-acoustics coupled with additional capture of eels to confirm spawning activity.
Reviewer 2 Report
General Remarks,
The present work is relevant and provides strong evidence of a seasonal aggregation of mature deep-sea eels individuals. Deep-sea ecological studies are complex and entail considerable investment costs.
In the introduction, the authors conclude that their findings clearly describe a spawning aggregation. This statement is inappropriate in the introduction and elsewhere. Anyway this kind of statement should belong to the conclusions.
In general, the methodological approach is sound but differs from each survey. This shows the opportunistic nature of the present work. Whit this point of view I’m not pretending to devalue it
In my opinion, a precautionary approach ensures robust knowledge. Despite this, authors consider that the examination of few individuals with an apparent maturing ovaries condition is irrefutable evidence of spawning aggregation. The Authors should take into account that life history traits studies or classical reproductive studies are based on an appropriate amount of individuals with a systematic sampling. To my knowledge, present work hasn’t provided robust evidence of spawning aggregation. No histological evidence verifies the spawning stage of the ovaries.
I understand the point of view of the authors and their intentions. I agree that a spawning aggregation is the most probable explanation. But, in my opinion, the Authors should reconsider the main finding and reconsider it as the most probable hipotesis.
This has implications in the manuscript (title), from the introduction to the conclusions.
Some comments:
Abstract
Line 15-15: “Apparent absence:”. This is a supposition, thus speculative
Line 17: d6ecades-long
Line 18: predict? How?
Line 21: “Unprecedent examination of a conservation-led recovery in the dep.” Again, this is speculative.
Keywords: Disatobranchus capensis. Did you mean Diastobranchus capensis?
Author Response
We are very pleased to have received three thoughtful reviews that each recommended only minor changes. These have enabled us to make some beneficial changes to our manuscript. The key issue for this paper, identified by all reviewers, is whether or not this aggregation can be confidently identified as a spawning aggregation. Each reviewer considered that our multiple-lines-of-evidence approach to interpreting the data was sound, that there was strong evidence for a spawning aggregation, but that it wasn’t conclusive.
To provide a conclusive account that fully characterised this aggregation and spawning activity would require multiple seasonal surveys using quantitative acoustics and capture of eels for full histological analysis. That kind of research effort can only be applied to species supporting industrial-scale fisheries (e.g. orange roughy), or to the highly valuable, enigmatic and elusive anguillid eels. We feel that our data set supports our conclusion, and that our manuscript finds the right balance between showing confidence in the data, and tempering this with a degree of caution. We also note that our approach is rather conservative alongside other studies of eel spawning aggregations cited in our Introduction that are based on considerably fewer data. Notwithstanding, we have made some adjustments to the text in the Abstract, Introduction and Discussion to emphasise that future validation of our conclusion is required. We have done this without watering-down the conclusion and have used less-speculative language as advised by the reviewers. Against this justification, we have not withdrawn the conclusion from the Abstract or Title.
Specifically, we have substantially modified the Introduction to avoid presenting the conclusion first, but have retained our conclusion of spawning elsewhere — including the title as justified above.
A full study of the reproductive biology within a dedicated survey of the aggregation would have been ideal, but this was not possible. We hope our work will stimulate such study in the future. Nonetheless, eels in running ripe condition is irrefutable evidence of imminent spawning activity, and their presence in an aggregation is strong evidence for that being where spawning occurs.
Line 15-15: “Apparent absence:”. This is a supposition, thus speculative. -- We understand the reviewers interpretation and have modified this language. It intended to convey that no other aggregation has been detected by extensive sampling and commercial fishing, but doesn’t rule out the existence of others.
Line 17: d6ecades-long -- Fixed
Line 18: predict? How? -- Based on the contrast between high fishing mortality and full protection.
Line 21: “Unprecedent examination of a conservation-led recovery in the dep.” Again, this is speculative. -- As above, this sentence has been modified
Reviewer 3 Report
This is a very interesting study exploring a population of basketwork eels, which highlights elements of their ecology and spatial distribution but also raises a critical eye over whether this is a spawning aggregation. The data have been fairly opportunistic, obtained utilising a range of techniques including video/photo observations along transects and acoustic surveys - but highlight a consistent pattern in the distribution of these fish. Morphological and behavioural analysis provides some interesting evidence to support the idea that these may be a spawning aggregation, although I feel the jury is still out here. Nevertheless, the data certainly provide a very useful insight particularly in a region which has previously been highly affected by fishing pressure.
The authors have dealt with many of the questions that arose from your data very well, and I don’t think many changes really need to be made to the manuscript as it stands. I’ve outlined some suggestions below:
- Section 2.3: Relative Abundance: To what extent could individual eels be counted more than once? Were eels sedentary or moving at the time of the transect e.g. could they be moving along with the transect?
- Section 2.4: How was euthanasia accomplished and the death of the fish assured?
- Methods General: You don't sufficiently describe methods of Analysis in the Methods section. Please do highlight the analytical processes. You do indicate that T-Tests were performed in the Methods but, in the Results, it seems like you have also conducted ANOVA, linear regression and Kruskal-Wallis tests. Indeed, it seems as though an ANOVA is reported for the data when the T test was suggested. So please do work on this for clarity. For these analyses please also provide the full statistical information (test statistic, df or n, p value) as well as these can be really informative together.
- Line 156: You indicate that you explored dentition here, but I don't think this is further described in the Results.
- Figure 6. Title: Please remove "showing" after (a) and (b)
- Line 228-229: You indicate slow moving and lack of flight responses in behaviour of these fish. Is this typical behaviour of the species? Or is it indicative of age/condition?
- Figure 8: You identify some regions in blue circles unlikely to be D. capensis; why is this the case? What features suggest the red-circled areas are this species? Is it proximity to the seamount or some other feature?
- Figure layout overall is unusual. Figure 8 should come after Figure 7, and Figure 9 may be useful either in Methods or in Results indicating some of the anatomical features of this species in an exploratory manner.
- In the paragraph beginning Line 344, you suggest this is a spawning aggregation to which eels migrate. However, you provide evidence (and describe this in your Discussion as well) that eels seem to be at this seamount year-round. Does that suggest these eels also spawn year round? You also mention that skin condition and behaviour is not consistent with spawning. Obviously there is compelling evidence of fish that are developing or at least approaching spawning capable and therefore likely to spawn soon but perhaps there needs to be some further recognition of the opposing evidence here. It may just be that our understanding of morphological and behavioural change in this species is insufficiently advanced to recognise a spawning aggregate, but I think caution is merited.
Author Response
We are very pleased to have received three thoughtful reviews that each recommended only minor changes. These have enabled us to make some beneficial changes to our manuscript. The key issue for this paper, identified by all reviewers, is whether or not this aggregation can be confidently identified as a spawning aggregation. Each reviewer considered that our multiple-lines-of-evidence approach to interpreting the data was sound, that there was strong evidence for a spawning aggregation, but that it wasn’t conclusive.
To provide a conclusive account that fully characterised this aggregation and spawning activity would require multiple seasonal surveys using quantitative acoustics and capture of eels for full histological analysis. That kind of research effort can only be applied to species supporting industrial-scale fisheries (e.g. orange roughy), or to the highly valuable, enigmatic and elusive anguillid eels. We feel that our data set supports our conclusion, and that our manuscript finds the right balance between showing confidence in the data, and tempering this with a degree of caution. We also note that our approach is rather conservative alongside other studies of eel spawning aggregations cited in our Introduction that are based on considerably fewer data. Notwithstanding, we have made some adjustments to the text in the Abstract, Introduction and Discussion to emphasise that future validation of our conclusion is required. We have done this without watering-down the conclusion and have used less-speculative language as advised by the reviewers. Against this justification, we have not withdrawn the conclusion from the Abstract or Title.
Specific points:
1. Section 2.3: Relative Abundance: To what extent could individual eels be counted more than once? Were eels sedentary or moving at the time of the transect e.g. could they be moving along with the transect?
An interesting question — but there was highly unlikely to be double-counting because there were gaps between images. In fact, the opposite was expected and reported, i.e. that our counts were underestimates.
2. Section 2.4: How was euthanasia accomplished and the death of the fish assured?
This is explained with an additional line of text (l. 147)
3. Methods General: You don't sufficiently describe methods of Analysis in the Methods section. Please do highlight the analytical processes. You do indicate that T-Tests were performed in the Methods but, in the Results, it seems like you have also conducted ANOVA, linear regression and Kruskal-Wallis tests. Indeed, it seems as though an ANOVA is reported for the data when the T test was suggested. So please do work on this for clarity. For these analyses please also provide the full statistical information (test statistic, df or n, p value) as well as these can be really informative together.
This is clarified with additional text in Section 2.4
4. Line 156: You indicate that you explored dentition here, but I don't think this is further described in the Results.
We have added a note to this effect (no differences detected in spawning fish)
5. Figure 6. Title: Please remove "showing" after (a) and (b)
Fixed
6. Line 228-229: You indicate slow moving and lack of flight responses in behaviour of these fish. Is this typical behaviour of the species? Or is it indicative of age/condition?
This is typical for all D. capensis seen by our tow cameras (usually lone individuals). A slight addition made to this sentence (l. 242).
7. Figure 8: You identify some regions in blue circles unlikely to be D. capensis; why is this the case? What features suggest the red-circled areas are this species? Is it proximity to the seamount or some other feature?
It is the intensity of the acoustic return coupled with proximity to the seamount. Additional text and annotation has been added to the figure legend to explain this better.
8. Figure layout overall is unusual. Figure 8 should come after Figure 7, and Figure 9 may be useful either in Methods or in Results indicating some of the anatomical features of this species in an exploratory manner.
Yes indeed. This appears to be partly a typesetting issue - and now noted. We have introduced text to section 3.2 to allow old Figure 9 to come earlier.
9. In the paragraph beginning Line 344, you suggest this is a spawning aggregation to which eels migrate. However, you provide evidence (and describe this in your Discussion as well) that eels seem to be at this seamount year-round. Does that suggest these eels also spawn year round? You also mention that skin condition and behaviour is not consistent with spawning. Obviously there is compelling evidence of fish that are developing or at least approaching spawning capable and therefore likely to spawn soon but perhaps there needs to be some further recognition of the opposing evidence here. It may just be that our understanding of morphological and behavioural change in this species is insufficiently advanced to recognise a spawning aggregate, but I think caution is merited.
Yes, this is an interesting point. We did note that there were many fewer eels in December and that those caught were not in spawning condition (although N= only 2). A similar phenomenon is seen in some other seamount-associated species such as orange roughy, i.e. particular seamounts become seasonal spawning sites, then most individuals disperse, but there is year-round presence at relatively high abundance compared to other adjacent sites. In the case of D. capensis, the year-round presence appears to remain relatively high compared to other adjacent seamounts, which may indicate protracted spawning activity. This will require considerable effort to determine. Nonetheless, we have made an additional notes about this uncertainty in the Discussion.
We made brief notes about external morphology and behaviour because differences were noted in spawning anguillid eels. This may, however, be a distraction in our account because there was no a priori expectation that this would also be the case with D. capensis. We have simply reported an absence of such changes.
Round 2
Reviewer 2 Report
I understand your position "To provide a conclusive account that fully characterised this aggregation and spawning activity would require multiple seasonal surveys using quantitative acoustics and capture of eels for full histological analysis. That kind of research effort can only be applied to species supporting industrial-scale fisheries (e.g. orange roughy), or to the highly valuable, enigmatic and elusive anguillid eels".
I disagree with this argument. There is a lot of information around life cycle or reproductive traits around none commercial deep-sea species . These studies have been based on research cruises/surveys. The present manuscript has relevant information and provide a valuable information but, as mention the authors, it is not conclusive around the nature of the eels aggregations, and the spawning aggregation has not been not determined (it is speculative).
Regards